# Neural representation of cytokines by vagal sensory neurons

Tomás S. Huerta[1], Adrian C. Chen[1,2], Saher Chaudhry[1], Aisling Tynan[1], Timothy Morgan[1], Kicheon Park [1], Richard Adamovich-Zeitlin[1,2], Bilal Haider[1], Jian Hua Li[1], Mitali Nagpal[1], Stavros Zanos [1,2,3], Valentin A. Pavlov[1,2,3], Michael Brines [1], Theodoros P. Zanos [1,2,3,4], Sangeeta S. Chavan [1,2,3], Kevin J. Tracey [1,2,3] ✉ & Eric H. Chang [1,2,3] ✉

The nervous system coordinates with the immune system to detect and respond to harmful stimuli. Inflammation is a universal response to injury and infection that involves the release of cytokines. While it is known that information about cytokines is transmitted from the body to the brain, how the nervous system encodes specific cytokines in the form of neural activity is not well understood. Using in vivo calcium imaging, we show that vagal sensory neurons within the nodose ganglia exhibit distinct real-time neuronal responses to inflammatory cytokines. Some neurons respond selectively to individual cytokines, while others encode multiple cytokines with distinct activity patterns. In male mice with induced colitis, inflammation increased the baseline activity of these neurons but decreased responsiveness to specific cytokines, reflecting altered neural excitability. Transcriptomic analysis of vagal ganglia from colitis mice revealed downregulation of cytokine signaling pathways, while neuronal activity pathways were upregulated. Thus, nodose ganglia neurons perform real-time encoding of cytokines at the first neural station in a body-brain axis, providing a new framework for studying the dynamic nature of neuroimmune communication.

Infection and injury initiate inflammation as a primary protective response through mechanisms refined over millions of years of evolution. Inflammation is modulated by counter-regulatory signals originating in the immune and nervous systems. One well-studied example of this neuroimmune control is the inflammatory reflex, in which vagal afferents are activated by cytokines and other factors. Brain neurons respond reflexively with signals to the body that inhibit cytokine production and attenuate inflammation[1–5]. This neural reflex regulation requires sensory afferent vagus nerve signals that travel to the brain, as transecting the nerve or blocking neural potentials disrupts cytokine-triggered systemic responses[6–8]. Vagal sensory neurons express receptors for cytokines and other immune mediators and

transmit cytokine-specific neural action potentials to the brain[9]. Moreover, recent findings have identified a body-brain neural circuit requiring vagus nerve signals that travel to specific brainstem nuclei to mediate pro- and anti-inflammatory signaling[10]. This work showed that specific vagal afferent neurons respond to pro-inflammatory and anti-inflammatory cytokines that then transmit inflammatory signals from the body to specific neurons in the brainstem nucleus tractus solitarius (NTS)[10]. However, it remains unclear whether individual vagal sensory neurons encode cytokines as specific neural signals in this body-brain neuroimmune axis.

The capacity to detect and precisely represent cytokine signals is important to ensure appropriate immune modulation by multiple

[1]Institute of Bioelectronic Medicine, Feinstein Institutes for Medical Research, Northwell Health, Manhasset NY, USA. [2]Donald and Barbara Zucker School of Medicine at Hofstra/Northwell, Hempstead NY, USA. [3]Elmezzi Graduate School of Molecular Medicine, Manhasset NY, USA. [4]Institute of Health System Science, Feinstein Institutes for Medical Research, Northwell Health, Manhasset NY, USA. ✉e-mail: kjtracey@northwell.edu; echang1@northwell.edu

neuroimmune reflex pathways and in the initiation of sickness behaviors[11,12]. The vagus nerve, cranial nerve X, widely innervates the viscera and mediates an extremely diverse range of interoceptive signals from the body to the brain[13–15], including cytokine-specific information[3,8,9,16]. Cytokines and the bacterial endotoxin lipopolysaccharide, administered systemically, are known to activate vagal sensory neurons[7,17]. Because the cell bodies of vagal sensory afferents reside in the vagal ganglia, we reasoned that the neural activity of individual neurons within these ganglia should be the first neural station along a body-brain neuroimmune axis that informs the brain of inflammatory events.

To examine this possibility, we performed in vivo calcium imaging of vagal ganglia in *Vglut2-GCaMP6f* mice. We observed that specific inflammatory cytokines are represented by distinct patterns of real-time neuronal responses in individual neurons of the nodose ganglia. Groups of individual nodose ganglia neurons were cytokine-selective, but other neurons functionally responded to multiple cytokines while maintaining distinct cytokine-specific patterns for each, indicating that immune signals have distinct neural representations. Nodose ganglia neuronal activity and cytokine-specific neuronal responses were both altered in mice with dextran sulfate sodium (DSS)-induced colitis, indicating that inflammation changes neural excitability.

## Results

### Nodose ganglia neurons have cytokine-specific neural responses

Peripheral sensory neurons respond to various stimuli and transmit information about organ function, energy metabolism, and immune status from the periphery to the brain[14,18]. The vagal ganglia complex that houses the cell bodies of the sensory vagus nerve is a merger between two developmentally distinct sets of neurons, the neural crest-derived jugular ganglia and placode-derived nodose ganglia[18]. Although anatomically fused in mice, the distinction between the populations is revealed by the expression of the transcription factors paired-like homeobox 2b (*Phox2B*) in nodose ganglia neurons and PR domain-containing member 12 (*Prdm12*) in jugular ganglion neurons (Fig. 1a). Within the ganglia, these pseudo-unipolar sensory neurons have bidirectional projections with one side innervating the viscera and the other terminating in the brainstem NTS. They are further differentiated by function and innervation pattern with jugular ganglia neurons carrying somatosensory information from the upper body and the nodose ganglia neurons carrying visceral signals from the major internal organs.

To provide insight into visceral cytokine signaling, we primarily imaged the larger inferior ganglionic structure of the nodose ganglia. We first confirmed that these were PHOX2B+ neurons that innervate the viscera. Histological assessment of the vagal ganglia subpopulations revealed that nodose ganglion cell bodies had significantly larger areas and diameters compared to jugular ganglion (Supplementary Fig. 1a, b, cell area, nodose vs jugular, 249.3 ± 7.2 vs 194.1 ± 12.1, **, $P < 0.003$; diameter, 17.5 ± 0.2 vs 15.6 ± 0.5, **, $P < 0.003$, Mann–Whitney test) confirming results from other groups[19,20]. We further differentiated the two vagal sensory neuron populations with a functional assessment by using the canonical transient receptor potential vanilloid 1 (TRPV1) activator capsaicin and the P2X receptor agonist α, β, methylene-ATP (αβmATP). Nodose ganglia, but not jugular ganglia, neurons express purinergic receptors and respond to αβmATP[21]. By applying these agonists directly to the vagus nerve, we found that capsaicin activated sensory neurons in both ganglia populations but that αβmATP only activated sensory neurons in the nodose ganglia (Supplementary Fig. 1c–e, JG vs NG, count, **, $P = 0.0022$, 0 ± 0 vs 20.67 ± 4.529, Mann–Whitney test). In our cytokine neuronal response experiments that follow, we are imaging nodose ganglia neurons that are PHOX2B+ and responsive to αβmATP.

Calcium imaging of the nodose ganglia reveals that the application of cytokines on the vagus nerve directly activates individual sensory neurons. By using a modified CaImAn pipeline and customized code[22], we extracted the calcium signals from detected regions-of-interest (ROIs) and plotted corresponding calcium traces (Fig. 1b). Representative calcium imaging traces show that the application of cytokines directly on the vagus nerve induces activity in multiple cells, in different locations of the nodose ganglion but without any discernable spatial pattern for cytokine responders (Fig. 1b). Cytokines were directly applied to the vagus nerve at the cervical level, immediately caudal to the vagal ganglia. To investigate if individual nodose ganglia neurons differentially encode cytokines, we applied three different cytokines directly on the vagus nerve: interleukin-1β (IL-1β), tumor necrosis factor (TNF), and interleukin-10 (IL-10) (Supplementary Movie 1).

Cytokine-specific responses were aligned from the start of the transient and averaged together across experiments demonstrating that these individual sensory neurons exhibit stereotyped responses to the same cytokine (Fig. 1c). Multiple response features were extracted from these cytokine-evoked responses, including amplitude, duration, rise slope, integral, number of peaks, and decay slope. We found that the maximum amplitude of cytokine-specific responses to IL-1β, TNF, and IL-10 were significantly different from each other (Fig. 1d, IL-1β vs TNF, 22.9 ± 2.6 vs 41.7 ± 11.3,**, $P = 0.002$; IL-1β vs IL-10, 22.9 ± 2.6 vs 73.3 ± 6.9, ****, $P < 0.0001$; TNF vs IL-10, 41.7 ± 3.4 vs 73.3 ± 6.9, ****, $P < 0.0001$, 1-way ANOVA). A comparison of additional extracted features demonstrated that distinct features of cytokine-evoked response profiles differed for each of the three cytokines tested (Supplementary Fig. 2). Next, we tested whether these cytokine-specific neuronal responses encoded any information about the cytokine intensity or concentration present, so we applied different concentrations of each cytokine (IL-1β: 100 ng/mL, 200 ng/mL, 400 ng/mL; TNF or IL-10: 25 ng/mL, 50 ng/mL, 100 ng/mL) to the vagus nerve. In these dose-response experiments, we did not find any evidence that cytokine concentration influenced the response profiles, indicating that the cytokine-specific responses are insensitive to cytokine concentration (Supplementary Fig. 3). Furthermore, when we applied cytokines to the proximal colon, an end organ innervated by the vagus nerve[23–25], we observed similarly stereotyped neural activity patterns for these three different cytokines, suggesting that there is a conserved representation for body-brain immune signaling at the end-organs (Supplementary Fig. 3).

### Nodose ganglia responses show that individual neurons can respond to multiple cytokines

To demonstrate the selectivity of cytokine-responsive subpopulations in the nodose ganglia, we applied two proinflammatory cytokines, IL-1β and TNF, sequentially onto the cervical vagus nerve while performing in vivo calcium imaging. We found four distinct subpopulations of nodose ganglia neurons in these experiments: IL-1β-specific, TNF-specific, multi-cytokine, and non-cytokine. A representative field-of-view image shows that these cytokine-responding neuronal subpopulations are widely distributed in the nodose ganglia, without any apparent topographic organization (Fig. 2a). The calcium transients that correspond to these subpopulations reveal that cytokine application triggers specific neural activity in the cytokine-responsive subpopulations compared to the baseline period (Fig. 2b). When the calcium transients are aggregated together as a heatmap (DFF normalized per row), the groupings of neuronal subpopulations become more clearly delineated (Fig. 2c). The proportion of total responders were categorized into four subpopulations that show a group of TNF responders (40.7%), followed by multi-cytokine (28.6%), then IL-1β responders (17.1%), with non-cytokine responsive neurons having the fewest calcium transients in these experiments (13.6%, Fig. 2d). Together, these results demonstrate that there are subpopulations of nodose ganglia neurons that can respond selectively to either IL-1β or TNF, or either inflammatory cytokine, with a fourth group of neurons

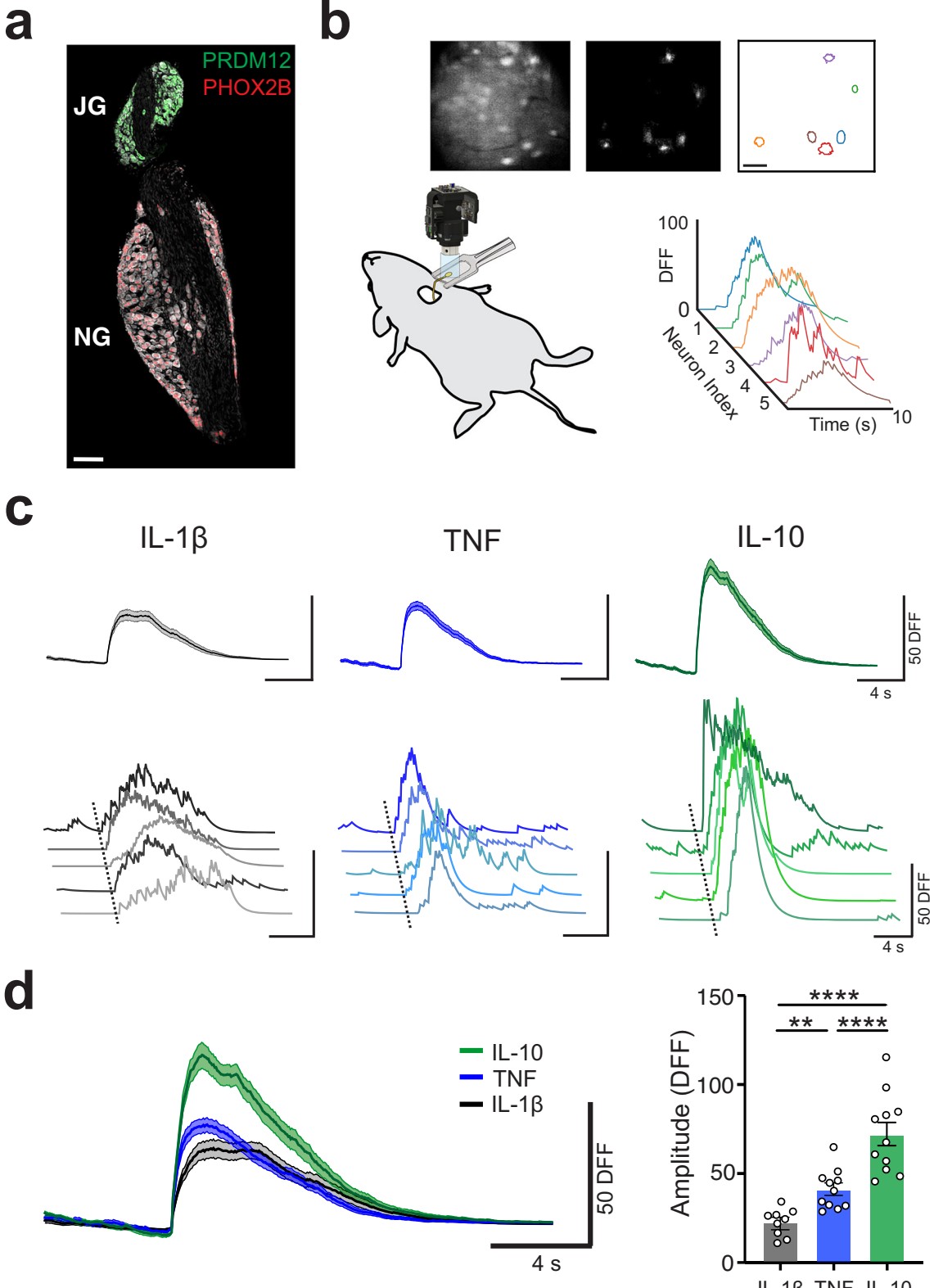

**Fig. 1 | Individual nodose ganglia sensory neurons respond in a cytokine-specific manner. a** PHOX2B and PRDM12 labeling reveals distinct subsets of placode-derived and neural crest-derived vagal sensory neurons in the jugular-nodose ganglionic complex. Scale bar, 100 μm. IHC was repeated independently 5 times. **b** Schematic of Miniscope in vivo calcium imaging of the mouse vagal ganglia. Top images show (left to right): raw fluorescence signal, DFF map, and a region-of-interest map showing active neurons; scale bar, 50 μm. Plot shows example individual calcium transient traces from six active neurons. **c** Distinct neuronal responses to specific cytokines (IL-1β, 200 ng/mL; TNF, 50 ng/mL, and IL-10, 50 ng/mL) administered to the vagus nerve. **d** Sensory neuron responses to cytokines are different from one another in their DFF amplitude (mean ± SEM, per mouse, $n = 9$, 10, 11, left to right, IL-1β vs TNF, *** $P = 0.0023$, IL-1β vs IL-10 and TNF vs IL-10, **** $P < 0.0001$, two-sided, one-way ANOVA, Tukey test corrected for multiple comparisons).

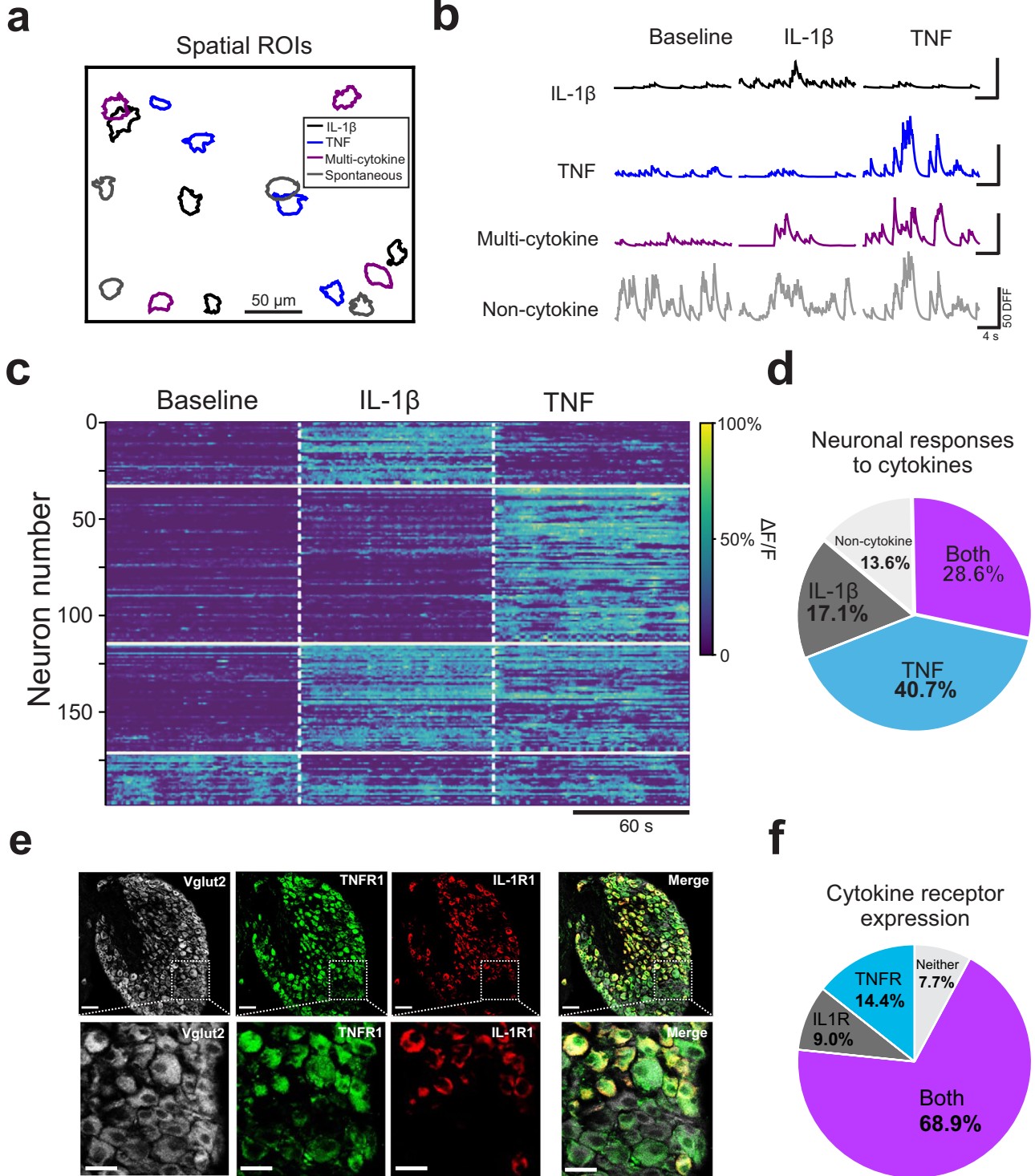

**Fig. 2 | Nodose ganglia population responses reveal subsets of cytokine-specific neurons while others respond to multiple cytokines. a** Representative field of view showing neurons that respond to specific cytokines. **b** Representative traces from individual sensory neurons responding to specific cytokines. **c** Heatmap demonstrating selective cytokine-specific responses in normalized neural activity index. White dotted lines indicate the time of cytokine application on the vagus nerve. **d** Pie chart depicting the proportion of total responsive neurons to specific cytokines, multiple cytokines or nonspecific activity. **e** Representative confocal microscopy images showing antibody labeling of IL-1R1 (red) and TNFR1 (green) cytokine receptors on nodose ganglia cell bodies. Scale bar, 50 μm; Zoomed inset scale bar, 25 μm. IHC was repeated independently in 5 mice, quantified over 2 sections each. **f** Quantification of nodose ganglion cell bodies labeled for IL-1R1 and TNFR1.

that is non-cytokine responsive. This non-cytokine group is likely to be a much larger proportion of the population than is shown here, as our analysis pipeline is designed to remove most of the neurons that are non-responsive to agonists.

Immunohistochemistry of the nodose ganglion showed that TNF receptor (TNFR1), and IL-1β receptor (IL-1R1) are found on nodose ganglia cell bodies (Fig. 2e and Supplementary Fig. 4a) and along the length of the vagus nerve (Supplementary Fig. 4b). Analysis of nodose

ganglia cytokine receptor labeling showed four subsets of *VGlut2-GCaMP6f*-positive sensory neurons expressing: TNFR1 (14.4%), IL-1R1 (9.0%), both IL-1R1 and TNFR1 (68.9%), or neither cytokine receptor (7.7%) (Fig. 2f). The immunochemistry data support the functional calcium imaging data that there are subpopulations of cytokine-specific responders, as well as individual sensory neurons that respond to multiple different cytokines.

## DSS-induced colitis in male mice increases cytokine levels in the colon and serum

To examine whether inflammation changes any nodose ganglia neuronal activity, we utilized the DSS-induced model of colitis, which is widely used due to its clinical and histopathological similarities to human inflammatory bowel disease (IBD). The DSS-colitis model involves a disruption of the gut epithelium, resulting in the stimulation of immune cells and elevations of cytokines and chemokines[26,27]. Male DSS-colitis mice displayed reduced body weight compared to control mice (normal drinking water) within 6 days of DSS exposure (Fig. 3a, *, $P = 0.0297$, Mixed-effects analysis). The severity of the DSS-colitis was tracked daily using a disease activity index (DAI) score, with the DSS-colitis mice demonstrating significantly increased DAI starting at day 3 and progressing until peak disease between day 7 and 8 (Fig. 3b, ***, $P < 0.0001$, Mixed-effects analysis). Post-mortem assessment of the colon revealed that DSS-colitis mice had a significant reduction of their colon length, a hallmark of colon inflammation (Fig. 3c, **, $P = 0.0028$, Mann–Whitney $U$ test). Colons were prepared into Swiss rolls and hematoxylin and eosin (H&E) stained to evaluate the degree of inflammation and tissue damage with a histological severity score. Colon tissue from the DSS-colitis mice had significantly higher histological severity scores compared to control mice (Fig. 3d, ***, $P < 0.0001$, Mann–Whitney $U$ test).

To assess the effect of systemic inflammation over multiple weeks, mice were examined for serum and colon cytokines at four different time points during the disease (day 2, 7, 14, and 21). Serum and colon cytokine and chemokine levels were quantified using a custom U-Plex Multiplex assay from Meso Scale Discovery. Serum levels of IL-1β are not significantly different at any time point (Fig. 3e). TNF levels were significantly elevated in DSS mice on days 7 and 14 (Fig. 3e, day 7, *, $P = 0.027$; day 14, *, $P = 0.037$, Mixed-effects analysis). IL-10 was significantly increased on day 7 (Fig. 3e, *, $P = 0.0246$, Mixed-effects analysis). In contrast with the serum results, colonic levels of IL-1β were elevated on days 7 and 14 (Fig. 3f, IL-1β colon, day 7, *, $P = 0.0257$; day 14, *, $P = 0.048$, Mixed-effects analysis). DSS colonic tissue samples also featured elevated levels of TNF and IL-10 on day 7 (Fig. 3f, day 7, TNF colon, **, $P = 0.009$; IL-10 colon, *, $P = 0.043$, Mixed-effects analysis). Other cytokines and chemokines, IL-6, CXCL1 and MCP-1 were significantly elevated in serum on day 7 (Fig. 3e, serum levels, IL-6 day 7, *, $P = 0.016$; CXCL1, day 7, **, $P = 0.0031$; MCP-1 day 7, *, $P < 0.031$, Mixed-effects analysis). IL6 and CXCL1 serum levels were elevated on day 14 (Fig. 3e, IL-6 serum, *, $P = 0.027$; CXCL1 serum, *, $P = 0.04$, Mixed-effects analysis). Similar results were found in homogenized colon samples (Fig. 3f, colon tissue, IL-6 day 7, *, $P = 0.039$; CXCL1 day 7, **, $P = 0.0049$; MCP-1 day 7, *, $P = 0.026$, Mixed-effects analysis).

## Inflammation increases the number of active nodose ganglia neurons at baseline, with lower amplitude responses

To test whether DSS-colitis inflammation alters the spontaneous baseline activity of nodose ganglia neurons, we examined their spontaneous neural activity at peak disease (DSS day 7) without applying cytokines. We found that the number of spontaneously active nodose ganglia neurons in baseline recordings from male mice with DSS-colitis was markedly increased at peak disease (Day 7), compared to control mice (Fig. 4a, b and Supplementary Movie 2). While there was an increased number of active sensory neurons in the DSS-colitis mice,

the amplitude of calcium transients from these mice was reduced (Fig. 4c, d, day 7, Control vs DSS, 42.66 ± 3.592 vs 29.08 ± 4.070, **, $P = 0.0260$, Mann–Whitney test). A comparison of the DSS and control group for additional extracted response features did not reveal any significant differences in other response features (Supplementary Fig. 2e).

To study the longer-term effects of DSS-colitis on nodose ganglia baseline neural activity, a separate cohort of male mice was followed for 21 days of DSS treatment. The disease activity score in the DSS group was significantly increased compared to controls (Fig. 4e, DAI, ***, $P < 0.001$, 2-way RM Mixed-effects Analysis). Similarly, assessment of their post-mortem colon lengths reveals that the DSS-colitis group had significantly shortened colon length compared to controls on days 7 and 14 (Fig. 4f, colon length, day 7, **, $P = 0.0078$; day 14, *, $P = 0.0274$, 2-way RM Mixed-effects Analysis). Calcium imaging analysis on this longitudinal cohort reveals that spontaneous transients observed in the DSS group were significantly lower than the controls on day 7 and day 14 (Fig. 4g, amplitude, day 7, *, $P = 0.0162$; day 14, *, $P = 0.0354$, 2-way RM Mixed-effects Analysis), which coincides with elevated scoring on the DAI. It is interesting to note that the disrupted neural activity at day 14 correlates with a shortened colon length in mice at this time point but not with the symptom-based disease activity scoring, which is nearly back to normal. This suggests the possibility that altered nodose ganglia neural activity may still persist through the resolution phase of colitis and may also be a more accurate indicator of end-organ health than disease activity scoring.

## DSS-colitis is associated with vagal ganglia transcriptomic changes in neuronal signaling and inflammatory signaling pathways

The elevation of several immune mediators in the serum and colon at peak DSS-colitis disease (Day 7) led us to ask whether this inflammation has any effects on the transcriptional profile of vagal sensory neurons. To this end, we performed deep, ultra-low input RNA sequencing to study the differential expression in vagal ganglia harvested at peak disease (Fig. 5a). RNA sequencing revealed that there was a marked upregulation of individual genes associated with neuronal signaling including *Scn10a, Syt7, Cacna1h, Cacna2d2, Grin3b, Grik4, Kcnk18, Kcnj12, Tpcn1, Dbh*, and a downregulation of genes associated with inflammatory signaling such as *S100a8, Ramp1, Ramp2, Tnfrsf12a, Tank, Tlr3, Il12a, Cxcl2, Ccl3, Ccl11* (Fig. 5b). There were also substantial changes in several gene ontology pathways (Biological Process) in terms of fold enrichment and false discovery rate (FDR) values. We found an upregulation in pathways linked to neuronal signaling, including notable increases in pathways associated with cation trans-membrane transport, neurotransmitter secretion, and the regulation of membrane potential (Fig. 5c). Additionally, in vagal ganglia from DSS-colitis mice, there was a specific downregulation of inflammatory signaling pathways, including those related to chronic inflammatory response, cell surface receptor signaling pathway via JAK-STAT, cellular response to TNF, and response to cytokine (Fig. 5d). These results indicate that DSS-colitis is associated with significant transcriptional changes in the vagal ganglia that selectively enhance or upregulate neuronal activity pathways, while simultaneously downregulating specific cytokine signaling pathways.

## Inflammation reduces sensory neuron responses in a cytokine-specific manner

Next, we wanted to examine whether inflammation alters nodose ganglia cytokine-specific responses. By performing in vivo calcium imaging on DSS-colitis mice at peak disease, we found a reduction in the amplitude of mean TNF and IL-10 responses in the DSS group when compared to controls (Fig. 6a). When responses to specific cytokines were compared to each other, IL-1β did not show clear changes in

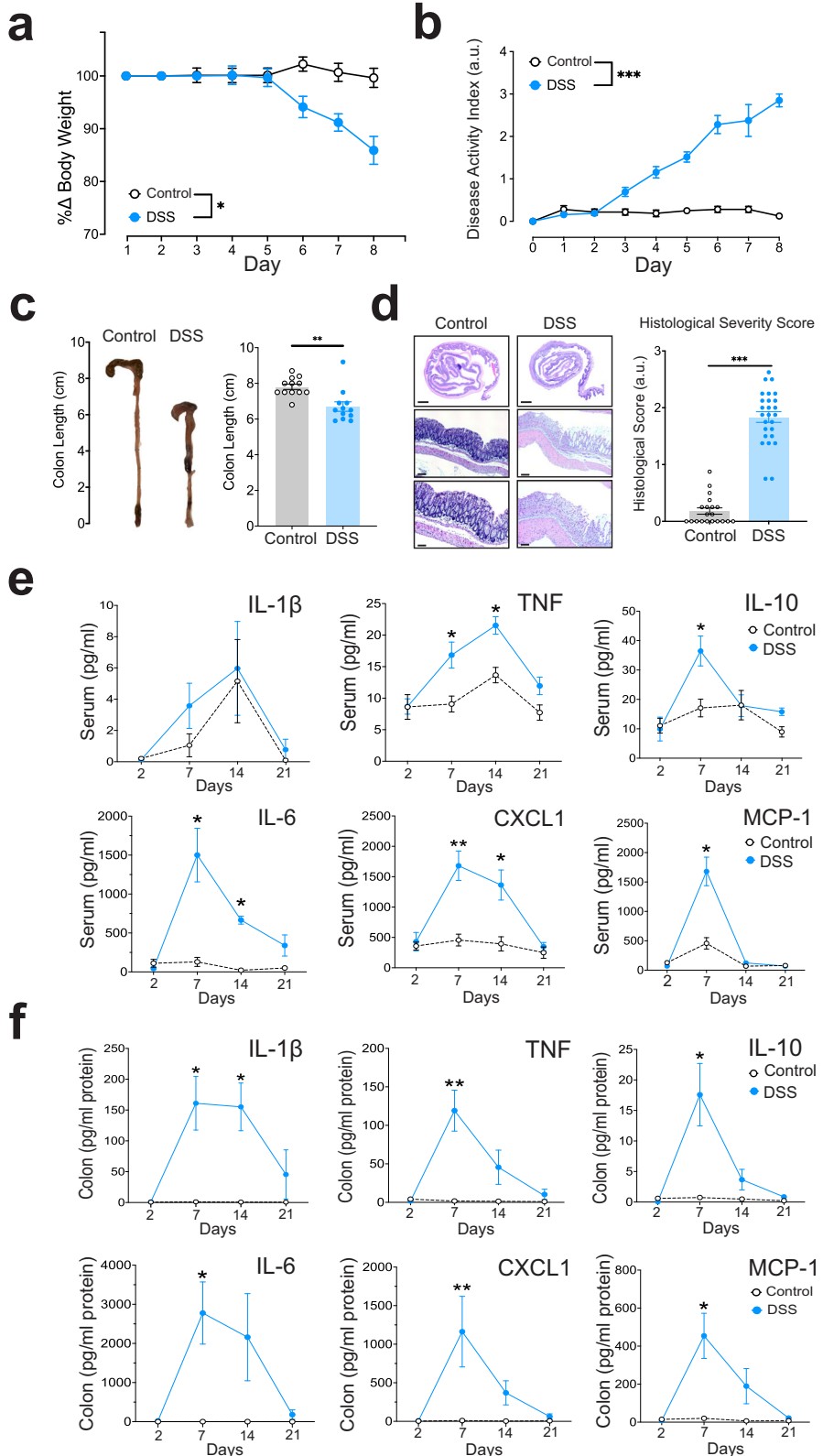

amplitude between DSS-colitis and control groups (Fig. 6b). In contrast, TNF-specific neuronal responses were significantly reduced in the DSS group (Fig. 6c, Control vs DSS, 56.62 ± 2.603 vs 44.27 ± 2.478, **, $P = 0.0062$, Mann–Whitney test). Similarly, IL-10-specific responses were diminished in the DSS group (Fig. 6d, Control vs DSS, 95.70 ± 8.923 vs 55.59 ± 3.264, ***, $P = 0.0003$, Mann–Whitney test). These results establish that DSS-colitis inflammation significantly

alters the cytokine-evoked activity of nodose ganglia neurons for specific cytokines, but not globally across all cytokines tested.

The marked changes in TNF and IL-10-specific signaling due to DSS-colitis provide evidence that inflammation in the body may lead to a reduction in the signal-to-noise at the individual neuron level. To examine this possibility, we used multidimensional clustering methods to plot points corresponding to each cytokine-evoked calcium

**Fig. 3 | DSS-colitis model damages gut tissue and increases serum and colon cytokines levels. a** DSS-induced colitis produces significant reduction in percent change of body weight (per mouse, *n* = 6, *, *P* = 0.0297, Mixed-effects analysis). **b** DSS-induced colitis produces an increase in disease activity score (*n* = 8 mice,***, *P* < 0.0001, Mixed-effects analysis). **c** Representative images of shortened colon length in DSS-colitis group, with associated quantification (Control, *n* = 12 mice, DSS-colitis, *n* = 14 mice, **, *P* = 0.0028, two-tailed Mann–Whitney test). **d** Histological assessment of colon tissue by hematoxylin and eosin (H&E) staining reveals severe histological changes in DSS-colitis colons (*n* = 20, 26, left to right, ***, *P* < 0.0001, two-tailed Mann–Whitney test). Scale bars for H&E images: top 1 mm, middle 100 μm, bottom 50 μm. **e** Colitis increases several serum cytokine levels at day 7 (peak disease, per mouse, *n* = 9, TNF day 7 *, *P* = 0.027, day *, *P* = 0.037; IL-10 *, *P* = 0.0246; IL-6 day 7, *, *P* = 0.016, day 14 *, *P* = 0.027; CXCL1 day 7, **, *P* = 0.0031, day 14 *, *P* = 0.04; MCP-1 day 7, *, *P* < 0.031, Mixed-effects analysis with Šidák correction). **f** DSS-colitis increases several cytokine levels in the colon at day 7 (peak disease, per mouse, *n* = 9, IL-1β day 7, *, *P* = 0.0257; day 14, *, *P* = 0.048; TNF day 7, **, *P* = 0.009; IL-10 day 7, *, *P* = 0.043; IL-6 day 7, *, *P* = 0.039; CXCL1 day 7, **, *P* = 0.0049; MCP-1 day 7, *, *P* = 0.026, Mixed-effects analysis with Šidák correction). All error bars represent ± SEM.

transient across the three most significantly different extracted features. Using this approach, we observed that cytokine responses in control mice were more distinguishable from each other than from DSS-colitis responses (Fig. 6e). Using a permutative Mann–Whitney test on a Calinski-Harabasz Index clustering algorithm, we found that the cytokine response distributions had significantly better clustering performance in controls compared to the DSS-colitis clusters (Control vs DSS, 47.11 vs 19.62, *P* = 0.0019). These data suggest that the DSS-colitis inflammatory condition alters the precision and separability of cytokine signals being communicated from the body to the brain.

## Discussion

Autonomic neural reflex circuits regulate organ function and tissue homeostasis through sensory afferent and motor efferent signaling. During infection and injury, sensory neurons are the initial line of defense to activate the appropriate neuroimmune regulatory circuits[28,29]. This innate regulatory mechanism exists in invertebrates with a primitive nervous system, such as *C. elegans*, and has been conserved throughout evolution[30,31]. Recent work has shown that different inflammatory challenges in the body can activate specific neuronal populations in the brain including via a vagal sensory afferent neuron pathway[10,11,32,33]. While the vagus nerve has long been implicated in body-brain signaling of immune signals[6,34], it was not known whether immune-responsive vagal sensory neurons encode information about the presence of specific cytokines in the body before transmitting this information to the brain. Here we show that nodose ganglia neurons directly respond to the cytokines TNF, IL-1β, and IL-10, which are principal elements of the innate immune inflammatory response. This specificity in the neural signals complements prior work showing that vagus nerve-recorded electrical activity responds to a range of physiological stimuli and immune challenges[9,14,35]. This peripheral encoding mechanism reflects a rapid neural route for the communication of specific immune information from the body to the brain via vagal sensory afferents that is distinct from the slower, humoral route that activates brainstem nuclei in the dorsal vagal complex, such as in area postrema[10,11]. The rapid communication of immune signals in the body to brainstem nuclei, such as the NTS[6,36], may enable innate reflexive mechanisms for protection and survival that require a faster time course than changes reflected in the bloodstream. These findings also fit into an emerging line of evidence that important sensory encoding and processing takes place outside of the central nervous system (CNS) in various peripheral ganglia or through organ-intrinsic sensory neurons, including the enteric nervous system of the gut[37,38], cardiac ganglia[39], and intra- and extra-pancreatic ganglia[40].

Vagal sensory neurons are a remarkably diverse and heterogeneous population, made of at least 24 genetically distinct transcriptomic clusters[18]. This genetic diversity is necessary due to the extraordinarily wide range of stimuli within the body that these sensory neurons must respond to and encode to maintain proper physiological homeostasis. Recent work on vagal sensory neurons has uncovered the various molecular mechanisms that allow these neurons to transmit and regulate the internal state of the body including signals related to cardiac function[41], respiration[42], feeding[43], and

digestion[44]. The role of the sensory vagus nerve in mediating aspects of immune system function has been known for decades[6,45]. However, only now are we beginning to decipher how immune mediators may be encoded in the body and how a few thousand sensory neurons per side in the murine vagal ganglia can be responsible for such a wide range of important interoceptive and immune functions[14,15].

There remain several open questions about how cytokines activate peripheral sensory neurons. The lack of sensitivity to cytokine concentration we observed in nodose ganglia neurons suggests a peripheral neural encoding system that prioritizes cytokine specificity over concentration (Supplementary Fig. 3a). Although the exact cytokine–receptor interactions that mediate these responses are unclear, prior work has shown that IL-1β can activate vagal afferents via transient receptor potential ankyrin 1 (TRPA1) ion channels[8]. Work from others has shown that inflammatory cytokines, such as IL-1β and TNF, sensitize peripheral sensory neurons by phosphorylation of voltage-gated sodium channels and TRP channels[46–48]. Moreover, type 2 cytokines, such as IL-4 and IL-13, can directly activate peripheral dorsal root ganglia neurons to drive itch[49] and sensitize lung-innervating nociceptors in allergic airway inflammation[50,51].

As evolutionarily conserved signaling proteins, cytokines can influence both immune and neuronal functions according to the site of action and the health status of the organism[30,52]. In the context of inflammation during DSS-induced colitis, we observed elevated levels of TNF that may underlie the increased baseline neuron activity observed (Fig. 4)[53], aligning with evidence that intestinal inflammation activates vagal afferents and the NTS[54–56]. During inflammation, neuropeptides such as calcitonin gene-related peptide play an important role in neuronal activation through its receptor RAMP1. In vagal ganglia from DSS-colitis mice, our transcriptomic data point to a downregulation of *Ramp1* and *Ramp2* alongside an upregulation of *Syt7*, a calcium sensor for neurotransmitter release[57], and the sodium channel *Scn10a* (Nav1.8) which plays an important role in membrane excitability (Fig. 5)[58]. We also identified changes in potassium channels (*Kcnj12*, *Kcnk18*), glutamate receptors (*Grik4*, *Grin3b*), and an upregulation of the gene encoding dopamine β-hydroxylase (*Dbh*), which has been recently linked to immune regulation[10]. Our functional data show that DSS-colitis increases the number of active nodose ganglia neurons at baseline, possibly owing to altered membrane excitability, with a reduction in the amplitude of cytokine-specific responses (TNF and IL-10). While IL-1β response amplitudes were not significantly decreased in DSS-colitis, there were decreases in other IL-1β neural response features (Supplementary Fig. 2). Together, these findings highlight only part of the complex neuro-immune interplay, which likely involves multiple sensory pathways and coordination between the nervous and immune systems to achieve homeostasis[48]. As other peripheral sensory neurons, including somatosensory DRGs, play important roles in neuroimmune responses, future work should aim to dissect the contributions of different sensory pathways to the temporal dynamics of inflammation such as the onset, duration, and resolution phases. It's possible that this type of immune information may integrated by the brainstem or higher-order CNS structures to engage neuroimmune reflexes or sickness behaviors[11,33,59].

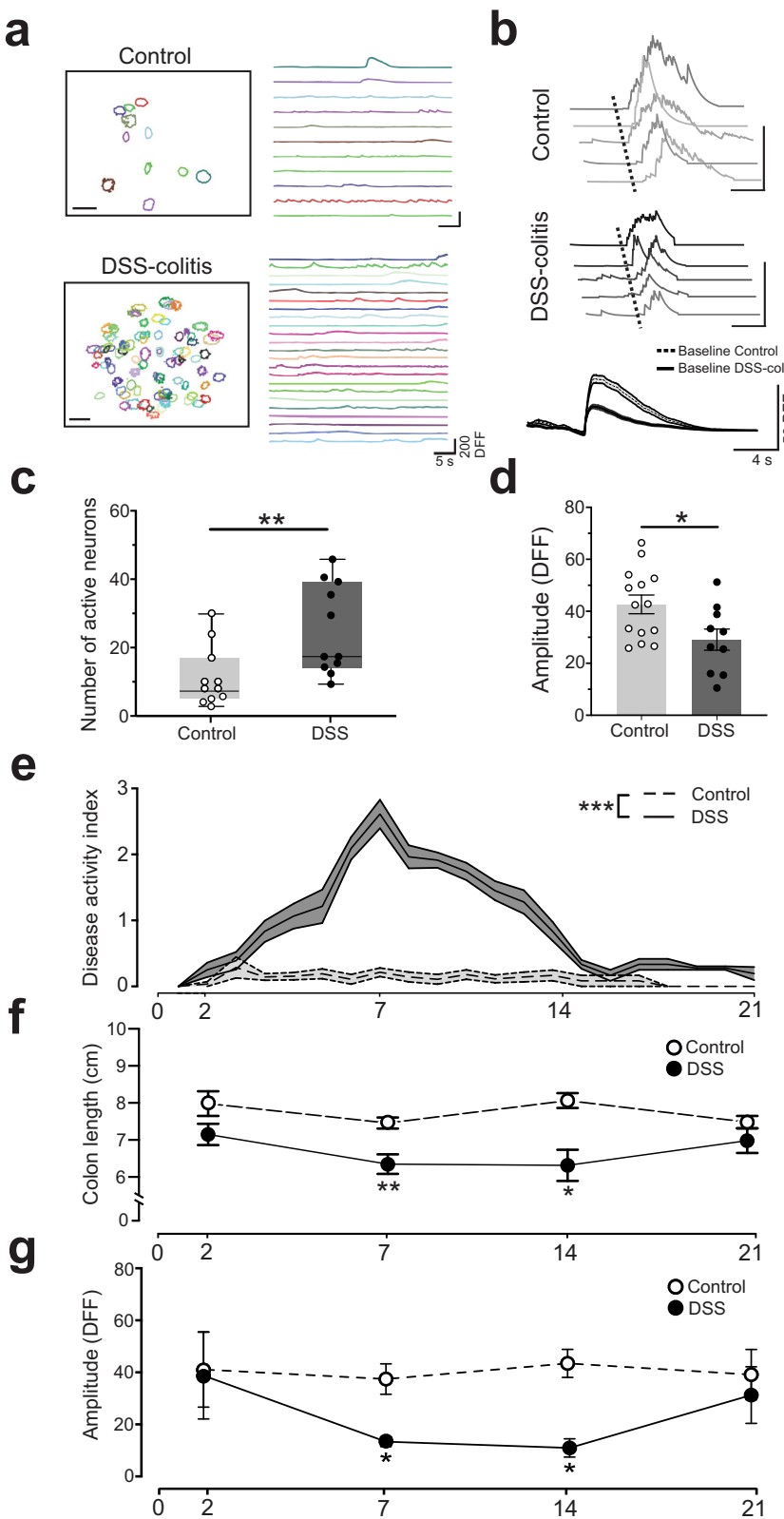

It is interesting to consider these results in the context of clinical studies for chronic inflammatory disorders of the gut, such as IBD. Visceral pain and hypersensitivity are often associated with conditions of gut inflammation, and increased vagal afferent pathway activity has been previously reported in animal models of colitis[60–62]. While it is unclear whether the increased number of active nodose ganglia neurons with reduced amplitude potentials in DSS-colitis mice reflects changes in sensing by gut-innervating afferents, this finding introduces the possibility that the sensory vagus nerve may be a viable target for neuromodulation-based therapies to treat IBD and other chronic inflammatory disorders[63]. In fact, vagus nerve stimulation has been shown to alleviate symptoms of IBD in both preclinical and clinical studies working via the inflammatory reflex to reduce cytokines and colonic inflammation[54,64,65]. In addition to the regulation of systemic

**Fig. 4 | Inflammatory state increases the number of active nodose ganglia neurons but decreases their response levels. a** Example ROI maps from a Control and DSS-colitis experiment, with sample traces from individual neurons during 50 s recording period. ROI map scale bar, 50 μm. **b** There were a higher number of spontaneously active nodose ganglia neurons in DSS-colitis mice at baseline (per mouse $n = 11$, whiskers indicate min/max value, bounds of box indicate 25% and 75% quartile, and line within box indicates median, ** $P = 0.0033$, two-tailed Mann–Whitney test). **c** Example spontaneous activity traces from individual neurons from Control and DSS-colitis mouse groups. **d** Comparison of baseline activity reveals a significant reduction in the amplitude of calcium transients during DSS-colitis (Control, $n = 14$ mice; DSS-colitis, $n = 10$ mice, * $P = 0.026$, two-tailed

Mann–Whitney test). **e** Daily measurements of disease activity index reveal increased disease severity in DSS mice (Control = 7 mice, DSS-colitis $n = 9$ mice, *** $P < 0.0001$, Mixed-effects analysis with Šidák correction). **f** Measurement of post-mortem colon length displays significant shortening on days 7 and 14 in the DSS group compared to controls (per mouse, $n = 12$, day 7 ** $P = 0.0078$, day 14 * $P = 0.0274$, Mixed-effects analysis with Šidák correction). **g** Analysis of baseline activity at several time points during disease progression reveals a significant reduction in the amplitude of spontaneous calcium transients in the DSS-colitis group at days 7 and 14 (per mouse, $n = 9$, day 7 * $P = 0.0162$, day 14 * $P = 0.0354$, Mixed-effects analysis with Šidák correction). All error bars represent ± SEM.

cytokines through the inflammatory reflex, there may also be other gut- and microbiome-specific mechanisms that contribute to the beneficial effects of vagus nerve stimulation in IBD including recently identified brain-body circuits that activate specific mucosal glands in the gut[66].

Our findings reveal that the sensing of cytokines through the vagus nerve in the body-brain axis is not a simple linear relay. Instead, nodose ganglia neurons create real-time, dynamic representations of specific cytokines before transmitting this information to the brain. We found that while some nodose ganglia neurons respond selectively to particular cytokines, others detect multiple cytokines yet maintain distinct activity patterns for each, suggesting a coding logic for immune signaling that is more complex than dedicated lines[67,68]. During DSS-induced colitis, these cytokine-specific neural activity signatures are altered, potentially disrupting immune signal representations in the body-brain axis, as seen in chronic inflammatory disorders and other diseases[69,70]. Together, these results offer a new framework for understanding how immune signals are dynamically encoded by the nervous system and how inflammation alters neural signaling.

## Methods
### Animals
Mice were housed on a 12:12 h reverse light/dark cycle at 22 °C and relative humidity of 30–70%. Water was available *ad libitum*. Experiments were carried out using adult mice (male and female) between 2 and 8 months of age. All experiments were performed under protocols approved by the Institutional Animal Care and Use Committee (IACUC protocol #2021-008) of the Feinstein Institutes for Medical Research (PHS Assurance #D16-00107) and conducted in strict adherence to the NIH Guide for the Care and Use of Laboratory Animals.

To monitor the neural activity of individual sensory neurons in vivo, we created VGLUT2–GCaMP6f mice by crossing homozygous VGLUT2-ires-Cre (Jax#028863) mice with homozygous Ai95D, also known as ROSA-GCaMP6f (Jax#028865) mice. In VGLUT2–GCaMP6f offspring, Cre recombinase expression is directed to excitatory glutamatergic neurons, where the floxed-STOP cassette for a GCaMP6 fast variant calcium indicator (GCaMP6f), inherited from the Ai95D parent, is trimmed. This leads to selective EGFP fluorescence in glutamatergic soma observed following calcium-binding, such as during neural activation.

**Nodose ganglion isolation and stabilization.** All surgical procedures were conducted using aseptic techniques. Animals were administered isoflurane anesthesia through a nose cone in the supine position (oxygen flow 1 L/min, isoflurane 1.75%). The mouse was positioned on a small animal physiological monitoring system, with integrated heating (Harvard Apparatus, Holliston, MA). An appropriate depth of anesthesia was assessed by toe pinch reflex. The hair was removed from the cervical region with Nair (Church & Dwight, Ewing Township, NJ), and then sterilized with 70% ethanol swabs.

A pair of micro-dissecting scissors (RS-5912SC Roboz, Gaithersburg, MD) was used to make a midline incision in the submandibular

area from the sternum to the chin. Four hemostat forceps (RS-7111, Roboz, Gaithersburg, MD) were used to pull the corners of the surgical window. A No.7 forceps (RS-5047) in the main hand and a No.7 Vessel Dilation forceps (RS-4927) in the offhand were used to tease apart the connective tissue above the intersection of the left vagus nerve and the hypoglossal nerve. Two retractors (18200-07 + 18200-09, Fine Science Tools, Foster City, CA) were used to pull the masseter muscles cranially and laterally. The nodose ganglion was then visualized under a branch of the carotid artery. A pair of micro-dissecting spring scissors (RS-5602) were used to cut the nodose on the coronal side. The nodose ganglion was gently pulled out of the surgical cavity and placed on the trachea. The vagus nerve was tracked down near the base of the neck, and isolated from the carotid bundle. Then, the vagus nerve was gently pulled through the trachea muscle and placed onto a custom mesh-covered retractor cut from a 40-μm EZFlow cell strainer (Foxx Life Sciences, Salem NH). The nodose ganglion was bathed in saline, and a UCLA Miniscope v4[71], with a coverslip mounted on the baseplate, was lowered onto the nodose ganglion using a stereotaxic frame. Prior to recording, the acquisition parameters were adjusted using the Miniscope DAQ recording software: 70 LED exposure, 3.5 gain, 20 FPS, 0.80 alpha, 0.10–0.15 beta.

**Utilizing the CaImAn pipeline.** We used the open-source analysis pipeline Calcium Imaging Analysis (CaImAn) to process our raw fluorescence data. CaImAn is a semi-automated software suite implemented in Python[72]. Prior to analysis with CaImAn, the FFV1 avi compression was converted to rawavi using FFmpeg[73]. All the files were then concatenated using the *concat* function in FFmpeg. FIJI was then used to crop the concatenated video around the nodose ganglion reduce black space from 600 × 600 pixels to ~350 × 350 pixels[74]. A python script was used to split the cropped concatenated file into 1000 frame tif files. The calcium imaging datasets were processed in CaImAn through a non-rigid motion correction algorithm (NoRM-Corre), followed by region of interest (ROI) detection by a constrained nonnegative matrix factorization for microEdoscope (CNMF_E) algorithm. Example videos were generated from DFF filtered datasets output from the CaImAn pipeline. Videos were denoised and brightness adjusted using Davinci Resolve with Neat Video plugin for enhanced presentation. Post-processing CaImAn pipeline output involved a custom Python package that automatically determined the integral for detected calcium transients. Only one response per calcium trace was analyzed. A cytokine-specific response was constrained to calcium transients that occurred within 100 s of cytokine application, in neurons that were not activate by prior application of the cytokine.

**DSS-induced model of colitis.** Colitis was induced by a supplement of 4% dextran sodium sulfate (DSS) in the water supply of mice (0216011080, MP Biomedicals, Irvine, CA). Only male mice were used in these experiments, as estrogen in females has been reported to reduce the severity of DSS-induced symptoms[75]. Mice were given 4% DSS in their water for 7 days. Animals were monitored daily for disease phenotypes including body weight change, diarrhea, and

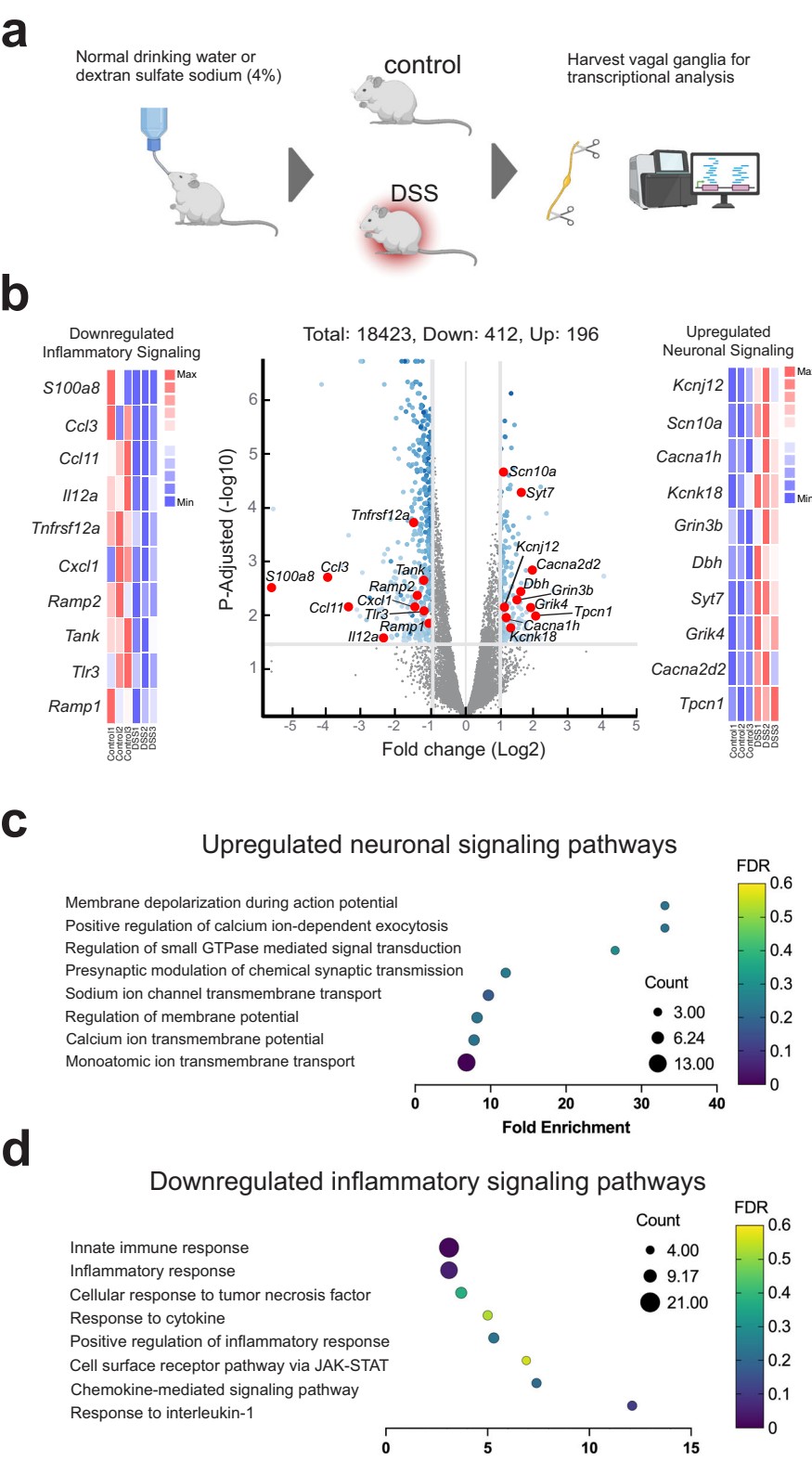

**Fig. 5 | Specific gene pathways related to neuronal and inflammatory signaling are altered in DSS-colitis vagal ganglia. a** Schematic showing the process for transcriptomic analyses of vagal ganglia from DSS-colitis mice. Created in BioRender. Chang, E. (2025) https://BioRender.com/poklo2e (**b**) Volcano plot showing individual transcriptomic changes in vagal ganglia from mice with DSS-colitis. Upregulated neuronal signaling associated genes (*Scn10a, Syt7, Cacna1h, Cacna2d2, Grin3b, Grik4, Kcnk18, Kcnj12, Tpcn1, Dbh*), downregulated inflammation

associated genes (*S100a8, Ramp1, Ramp2, Tnfrsf12a, Tank, Tlr3, Il12a, Cxcl2, Ccl3, Ccl11*). P-adjusted value was calculated using Benjamini-Hochberg correction, significance threshold was FDR < 0.05. **c** Weighted dot plot of a selection of significantly enriched biological pathways reveals upregulated neuronal pathways at peak colitis. **d** Weighted dot plot shows selected sets of downregulated genes associated with inflammatory signaling.

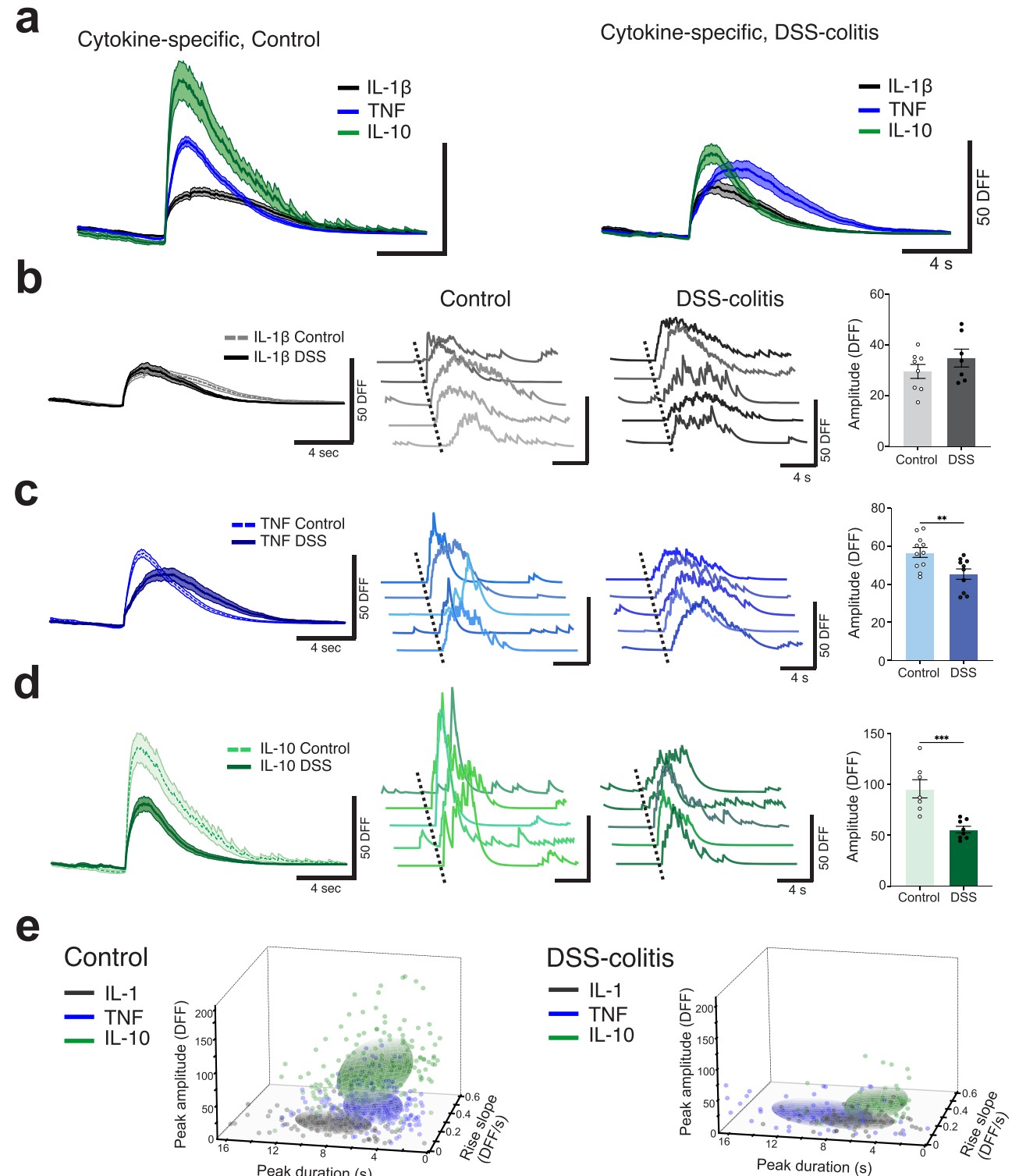

**Fig. 6 | Inflammation reduces sensory neuron responses in a cytokine-specific manner. a** Comparison of mean traces of sensory neuron responses demonstrates that DSS-induced colitis alters the responses to specific cytokines (TNF and IL-10) but not others (IL-1β). **b** Mean traces, representative traces and quantifications reveal that IL-1β responses do not change significantly during colitis (per mouse, $n = 8, 7$). **c** Mean traces, representative traces and quantifications reveal that TNF response amplitudes are significantly reduced during colitis (per mouse, $n = 11, 10$,

$**, P = 0.0062$, Mann–Whitney test). **d** Mean traces, representative traces and quantifications reveal that IL-10 response amplitudes are significantly reduced during colitis (per mouse, $n = 7, 8$, $***, P = 0.0003$, Mann–Whitney test). **e** Differentiating neural responses to the three cytokines in a multi-dimensional space. Response clusters from nodose ganglia neurons in DSS-colitis mice have significantly reduced separability (per response, IL-1β, TNF, IL-10, Control, $n = 153, 264, 223$, DSS: $n = 78, 83, 80$, respectively). All error bars represent ± SEM.

hematochezia (bloody stool). The severity of the disease was scored as the disease activity index (DAI score). When animals reached peak disease (≥3 DAI score, day 7–8), the mice were anesthetized and subjected to calcium imaging of the nodose ganglion. A separate cohort of mice were assessed at several timepoints along the progression of DSS-induced colitis (Day 2, Day 14, and Day 21). Key resources including chemicals and antibodies are listed in Supplementary Table 1.

**Application of solutions directly on the vagus nerve.** The cytokines TNF, IL-1β, and IL-10 were generated in-house and diluted to the μg/mL dose. Pilot dose-response experiments were conducted to determine the minimal cytokine dose necessary to induce cytokine activity from direct application of the cytokine to the vagus nerve. The concentrations for each cytokine were: TNF (50 ng/mL), IL-1β (200 ng/mL), and IL-10 (50 ng/mL). Solutions were directly applied to the exposed vagus nerve as 15 μL droplets using a 20 μL micropipette (3123000039 Eppendorf Research Plus, Enfield, CT). In control experiments, saline was applied to the vagus nerve which activated a subset of nodose ganglia neurons that were independent of the cytokine-responders and likely to be mechanosensitive sensory neurons.

**Post-mortem assessment of harvested tissue and cytokine level measurements.** After nodose ganglion recordings mice were euthanized and nodose ganglion, blood serum and colon were harvested for post-mortem assessment. Nodose ganglion samples were subjected to either immunohistochemistry or bulk RNA-seq transcriptomic analysis. Colon samples were measured to assess severity of colitis-associated shortening. Colon tissue samples were homogenized using a Bullet Blender (BBY24M, Next Advance, Troy, NY). The cytokine levels of the homogenized tissue samples and blood serum samples were assessed with a custom U-plex multiplex immunoassay targeting the following biomarkers: TNF, IL-1β, IL-6, IL-10, IL-23, KC/GRO, MCP1 (U-Plex Custom Biomarker Group 1, K15069M-2, Meso Scale Discovery, Rockville, MD). A subset of colon samples were stained for H&E histological assessment (AML Laboratories, St Augustine, FL).

**Vagal ganglia isolation for bulk RNA-seq analysis.** Mice were given 4% Dextran Sodium Sulfate or normal drinking water for 7 days to induce DSS-Colitis ($n = 6$/group). On Day 7 or 8, mice were euthanized and the left and right nodose ganglia harvested in ice-cold HBSS. Nodose ganglion tissue was rinsed with HBSS buffer (+ $Ca^{2+}$ and $Mg^{2+}$). The buffer was removed and replaced with 1 mL of warm NB medium (1% pen/strep, 1× glutmax, 50 ng/mL nerve growth factor, 2% B27, in Neurobasal Media). 10 uL of collagenase/dispase (100 mg/mL) was added to solution, which was then incubated at 37 °C on a rotating shaker for 90 min. After incubation, the tissue was allowed to settle, the supernatant was removed, and the tissue was rinsed twice with HBSS. The tissue was then triturated in 1 mL of fresh warm NB media using a glass Pasteur pipette until the intact ganglion dissociated. The dissociated cells were filtered through a 70 μM filter. The cells were then spun at 130 RCF for 5 min and the pellet was resuspended in 300 μL NB media. The cell suspension was layered onto 1 mL of 15% filtered BSA in HBSS and centrifuged at 130 RCF for 20 min without brake. The supernatant was removed, and the pellet was then resuspended in 100 μL of PBS and centrifuged at 300 RCF for 10 min. The supernatant was completely removed, then the pellet was resuspended in 5 μL of PBS and flash-frozen in liquid nitrogen. Frozen samples were sent to SingulOmics (Bronx, NY) for bulk RNA-sequencing using Takara SMART-Seq v4 Ultra Low Input RNA Kit for Sequencing.

**Bulk RNA transcriptomic analysis.** To perform pathway analysis of differentially expressed genes (DEGs) from bulk RNA sequencing data of DSS versus control nodose ganglia, the data was processed using the Basepair platform (New York, NY). Basepair generated an initial list of DEGs based on statistical analysis of the RNA sequencing data. This list was subsequently filtered to include only genes with an absolute log2 fold change >0.5 and a $p$-value < 0.05. The filtered DEGs were further categorized into upregulated and downregulated groups depending on whether the log2 fold change was positive or negative in the DSS vs. control comparison. The filtered DEG lists were analyzed using the Database for Annotation, Visualization, and Integrated Discovery (DAVID) tool[76,77], employing an over-representation analysis (ORA) approach. ORA identifies enriched biological pathways by focusing on genes that meet specific significance thresholds, such as those included in the filtered DEG lists. During the analysis, *Mus musculus* was specified to map mouse gene identifiers to their respective annotation terms. Gene Ontology (GO) Biological Process enrichment analysis was performed separately for the upregulated and downregulated DEG lists to identify significantly enriched biological pathways. The results were interpreted to highlight pathways relevant to the observed differential expression patterns.

**Multidimensional cluster analysis.** Five features (amplitude, duration, rise slope, decay slope, and integral) were extracted from cytokine-evoked calcium transients in both Control and DSS conditions. The five features were plotted as points in multidimensional space grouped by their respective cytokine. After outliers were removed with Tukey's fence (Q1/Q3 ± 1.5 × IQR), preprocessing was done by iterative nearest neighbor analysis dropping the 25% most similar point between cytokine groups. After preprocessing, cluster analysis (Calinski-Harabasz Index) was conducted across the three cytokine group for the Control and DSS conditions. Further, the quality of the cluster grouping was compared across the control and DSS-colitis condition with a permutative Mann–Whitney test to determine the significance of differences between respective clustering metrics[78]. 3D plots of the most significantly different extracted features (amplitude, duration, rise slope) were used to visually display the separability of the different groups between the control and DSS condition. 1 SD ellipsoid clouds were added to each cytokine-evoked group to aid the visualization of the categories. The colored gradient was applied only for improved visual acuity.

**Immunohistochemistry.** Vagal ganglia were excised and fixed in 4% PFA solution at 4 °C overnight and then transferred to 30% sucrose solution. Following embedding in (optimal cutting temperature) OCT compound, tissue sections (10–12 μm) were prepared on a Leica CM1850 cryostat. Following fixation, tissues were permeabilized with 0.1% Triton X-100 PBS solution and incubated in a blocking solution consisting of 10% Normal donkey serum (Southern Biotech, 0030-01), 10% Donkey Anti-Mouse Fab Fragments, and 0.1% Triton X-100 in 1× PBS for one hour at room temperature. Tissues were incubated with the following primary antibodies diluted in blocking buffer overnight at 4 °C: rabbit IL1RA (Abcam, Ab124962), mouse TNFR1 or rabbit TNFR1 (Proteintech, 60192-1-Ig), rat IL10RA (Abcam, Ab33738) or rabbit IL10RA (ThermoFisher, PA5-109852), PHOX2B (Abcam, Ab183741) or rabbit PHOX2B Alexa Fluor 647 (Abcam, Ab311130), Rabbit PRDM12 (EMD Millipore, ABE95), and Chicken Anti-GFP (Aves Labs, GFP-1010). Next, tissues were rinsed three times with 1× PBS-Triton X-100 (0.1 M PBS and 0.1% Triton X-100) and incubated with the following secondary antibodies diluted in blocking buffer: Donkey Anti-Rabbit Alexa Fluor 405, Donkey Anti-Rabbit Alexa Fluor 568, Donkey Anti-Rabbit 647, Donkey Anti-Rat Alexa Fluor 647, and Donkey Anti-Chicken (ThermoFisher, A78948). Some vagal ganglia tissue was labeled with Fluorescent Nissl (NeuroTrace 435/455; ThermoFisher, N21479). To eliminate high background and nonspecific labeling from same species labeling, mouse antibodies were conjugated with FlexAble Coralite 555. Similarly, multiplexing using rabbit antibodies was conjugated with FlexAble CoraLite 555 or 647 kits and stained sequentially to

prevent cross-reactivity of same-species antibodies. Lastly, tissues were rinsed with 0.1% Triton X-100 in 1× PBS and cover slipped with Fluoromount Mounting Medium (Southern Biotech, 0100-01). The specificity of anti-rabbit, anti-mouse, and anti-rat antibodies were validated using appropriate isotype antibodies as negative controls. Fluorescent images were captured with a Zeiss LSM 880 confocal microscope. Cell counting was conducted through single channel annotation in Zen Microscopy Software (Blue Edition, Carl Zeiss Microscopy, White Plains, NY).

For colon immunohistochemistry, colons were excised and fixed in 4% PFA overnight and cryopreserved in 30% sucrose solution at 4 °C. Tissues were embedded in OCT (optimal cutting temperature) and snap-frozen in dry ice. Radial sections were cut into 30 μm slices with a Leica CM1850 cryostat. Sections were permeabilized with 0.1% Triton X-100 PBS solution and incubated in blocking solution consisting of 10% normal donkey serum (Southern Biotech, 0030-01) and 0.1% Triton X-100 in 1× PBS for one hour at room temperature. Next, tissues were incubated with rabbit Anti-Beta Tubulin III (Abcam, Ab18207) and Goat Anti-GFP (Genetex, GTX26673) overnight at 4 C. Next, tissues were rinsed three times with 1× PBS-Triton X-100 (0.1 M PBS, 0.1% Triton X-100) and incubated with Donkey Anti-Rabbit Alexa Fluor 647 and Donkey Anti-Goat 488 for 1 h at room temperature. Tissues were rinsed with 0.1% Triton X-100 in 1× PBS and counterstained with DAPI. Autofluorescence was quenched using a Vector TrueView Autofluorescence Quenching Kit (Vector Laboratories, SP-8400). Sections were cover slipped with Fluoromount Mounting Medium (Southern Biotech, 0100-01) and imaged using an ECHO spinning disk confocal microscope. Images were processed using Zen Blue and Image-Pro 10.1. Antibodies and reagents used can also be found in Supplementary Table 1.

**Statistics and reproducibility.** For this study, we used animal numbers ranging from 3–14 mice per experimental group. A minimum of 10 calcium traces per sample were collected and averaged per animal. A minimum of $n = 3$ per group was used in dose-response experiments for different cytokine concentrations, longitudinal colitis experiments at various time points (days 2, 14, and 21), and RNA-sequencing transcriptomic analysis following quality control. For comparison of cytokine-specific responses to colon administration, group sizes of 4–6 mice were used. For comparisons of cytokine-specific responses to direct application to the nerve, 9–14 mice per group were used. For immunohistochemistry analysis, five replicates were used, with two fields of view assessed per sample. Comparisons between two groups at the same time point, such as for baseline neural activity, were analyzed with a two-tailed Mann–Whitney $U$ test. Comparisons between three or more groups at the same time point, as seen with neural responses across three cytokines, were analyzed using one-way ANOVA with Tukey correction for multiple comparisons. Comparison between two groups over multiple time points, including colitis data for body weight, colon length, and neural activity, were analyzed with Mixed-Effects ANOVA; a Šidák correction was used when making multiple comparisons (serum and tissue cytokine levels). Transcriptomic data was adjusted with a Benjamini-Hochberg correction. Data plots were generated using Python or with Prism 10 (GraphPad). All data in column graphs are presented as mean ± SEM, unless otherwise noted. Whisker-box plot features were min/max for whiskers, 25%–75% for box bounds, and median for line within box.

**Reporting summary**
Further information on research design is available in the Nature Portfolio Reporting Summary linked to this article.

## Data availability
Source data are provided with this paper. RNA-sequencing data are available at GEO (accession number: GSE294447). Source data are provided with this paper.

## Code availability
Jupyter notebooks and relevant code associated with this study are available at the following link: https://github.com/Eric-H-Chang/calcium-imaging-cytokines.

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

## Acknowledgements

The authors thank Drs. I.M. Chiu, I.T. Mughrabi, and D.F. Nixon for valuable feedback on the manuscript. They also thank members of the Laboratory of Biomedical Science for constructive comments throughout the project. Figure 5a and part of Supplementary Fig. 3 were created with BioRender. Miniscope schematic images in Fig. 1b and Supplementary Fig. 3 are from the UCLA Miniscope project. This work was supported by funds from the following NIH grants: R01GM143362 (EHC), R35GM118182 (KJT), and R01GM132672 (SSC).

## Author contributions

T.S.H. and E.H.C. designed the experiments. T.S.H., A.C.C., S.C., A.T., K.P. and M.N. performed experiments and acquired data. T.S.H., S.C. T.M., R.A., B.H., A.T., J.L., T.P.Z. and E.H.C. analyzed data. T.S.H., K.J.T. and E.H.C. wrote the manuscript. S.Z., V.A.P., M.B. and S.S.C. provided input and edited the manuscript. E.H.C. conceived the study and supervised the work.

## Competing interests

The authors declare no competing interests.
