## [Transparent Peer Review file · Nature Communications]

Neural representation of cytokines by vagal sensory neurons

Corresponding Author: Dr Eric Chang

Version 0:

Reviewer comments:

Reviewer #1

(Remarks to the Author)

In the present study, Huerta et al. investigated the neuronal responses to immune signals generated by nodose ganglion neurons in mice. Using in vivo calcium imaging, the authors showed that nodose vagal sensory neurons display distinct responses to pro-inflammatory (TNF, IL-1 β) and anti-inflammatory (IL-10) cytokines. These responses give rise to a population coding that involves both fine-tuned and broadly-tuned neurons. Using a dextran sulfate sodium (DSS)-induced colitis model, the authors show that inflammation alters nodose ganglion neural excitability. The authors concluded that nodose ganglia neurons perform real-time cytokine encoding at relay stations upstream of the brainstem. This paper presents a very interesting study in which the real-time responses of vagal neurons to cytokines were demonstrated. This is an important conceptual advance, as the data imply that rapid, real-time information on cytokine levels is conveyed to the brain, where counter-regulatory responses are presumably generated. Overall, the study is well-performed, timely, and well presented.

However, I am intrigued by the following. The data in Figure 1E clearly show no significant change in the amplitude of cytokine-evoked responses after the application of different concentrations of the same cytokine. However, DSS induced important changes in the excitability of nodose neurons. As DSS presumably changes cytokine levels, I expected concentration-dependent responses in nodose neurons. How did the authors interpret these results?

In addition, limited information on the transcriptomics data were provided. Specifically, do cytokine-responsive neurons also respond to enteroendocrine hormone signals? Are nodose neurons responsive to cytokines specifically targeting certain organs, such as the GI tract, as the DSS data suggest? How do nodose neurons sense circulating cytokines? Please comment.

Reviewer #2

(Remarks to the Author)

The authors demonstrated a real-time neural response to pro- and anti-inflammatory cytokines within the nodose ganglia and their selectivity of responses to these cytokines. The authors also showed that neurons in nodose ganglia are activated but their signaling capacity to specific cytokine was decreased under DSS-induced colitis. This study is very important in that vagal sensory neurons could encode cytokines as neural signals under disease. However, there still remains unclear how encoding cytokine-dependent signaling in VG neurons could impact systemic diseases in VG-dominated organs. Here are some comments from the reviewer.

#1. The authors have shown that vagal nerves respond to cytokine stimulation in terms of DFF amplitude and the pattern of DFF differed depending on the types of cytokines. However, it is vague what functional outcomes could be brought about after cytokine stimulation onto the vagal nerves. Without this functional assay, it is unclear how responses to cytokines in vagal nerve could impact on DSS-induced colitis. One suggestion could be evaluating the production of neuropeptides after cytokine exposure onto the vagal nerve and also testing if there is any cytokine-dependent differences.

#2. The author found more active nodose ganglia neurons under DSS-condition at day7 (Figure 4B). In the meantime, they also found that these nodose ganglia neurons are less sensitive to cytokine stimulation. The authors explain this discrepancy could reflect the desensitization of nodose ganglia neurons. However, a body of studies have proven protective roles of VG in reducing systemic inflammation including DSS-induced colitis. If VG becomes desensitized during DSS colitis and less responsive to cytokines, how could VG modulate disease through vagus nerves stimulation.

#3. The authors showed neural responses to IL-1 β and TNF α in Figure.2 but could they also show any data on the stimulation with IL-10 and IL-10R expression in VG? The reviewer is interested in IL-10 data because the authors have shown that VG neurons are more sensitive to IL-10 compared to other cytokines according to DFF amplitude (Figure.1 D,E). Do IL-10 responsive (or IL-10R expression) subset of nodose neurons possess more distinctive features?

#4. Colon is innervated from DRGs as well as VG, and DRGs could also encode cytokine signals. Then, to what extent vagal cytokine sensing is involved in this study? Are enteric DRG neurons could also show similar response to IL-1 β , TNF α and IL-10?

Reviewer #3

(Remarks to the Author)

Huerta used in vivo imaging of nodose ganglia in Vglut2-GCaMP6f mice to investigate the effects of pro-inflammatory cytokines (TNF, IL-1 β) and the anti-inflammatory cytokine IL-10 on neuronal responses in the nodose ganglia. Their findings suggest that some nodose ganglia neurons exhibit cytokine-selective responses, as indicated by increased GCaMP6f signals. Furthermore, the authors observed cytokine-specific patterns of activity, though the biological significance of these diverse responses remains unclear.

The study also assessed neuronal activity in the nodose ganglia using a dextran sulfate sodium (DSS)-induced colitis model. DSS-induced inflammation increased the number of active nodose ganglia neurons at baseline but reduced their signaling capacity and responsiveness to specific cytokines, indicating altered neural excitability. Transcriptomic analysis of vagal ganglia from DSS-treated mice revealed a down-regulation of cytokine signaling pathways, alongside an up-regulation of neuronal activity pathways. However, the mechanistic connections between these transcriptomic changes and functional outcomes were not investigated, limiting the scope of this study to an observational study, due to limited mechanistic insight, leaving the broader significance of the findings unresolved.

A primary concern for this reviewer is the reliability of the detected signals. Careful examination of the supplemental movie and the representative image in Fig. 1 suggests a low detection threshold. On page 4, the authors state that cytokine concentration influenced response profiles, yet they also imply that the cytokine-specific responses were insensitive to concentration. This inconsistency raises concerns. Based on Supplemental Movie 1, the low detection threshold—likely due to a low signal-to-noise ratio—may have compromised the accuracy of the results and, consequently, the interpretation of the data.

Additional comments: In Fig 1D the signal amplitude for IL1B, TNF and IL10 on average are around 25, 50 and 75, but for the dose response experiment in 1E the average signal amplitude is between 50 to 60 – what dose they used for D? why the differences are gone? Are the Ns relating to the activated number of cells or combined # of slice plus activate cell number Please include sample images for each condition- it is very important to see the images before they are processed for analysis.

What is the mechanistic link between Data shown in Fig 5 (Transcriptomic analysis of DSS-colitis vagal ganglia), and data shown in Figure 1-4?

Version 1:

Reviewer comments:

Reviewer #1

(Remarks to the Author)

This is an improved version of an elegant manuscript reporting on neuronal sensing mechanisms of immune signals. The authors have in my view addressed the concerns to satisfaction, so that the present version constitutes an important and timely contribution to the field.

Reviewer #2

(Remarks to the Author)

The authors have sufficiently addressed all concerns.

Reviewer #3

(Remarks to the Author)

The signal-to-noise ratio of the best representative raw data for each of the three cytokines tested further supports a near detection limit for the Miniscope images before and after IL1B, TNF and IL10. Aside from this concern, the authors have adequately revised the manuscript and addressed this reviewer's comments. I enjoyed reading this paper.

Thank you for reviewing our manuscript and spending the time to provide detailed and insightful comments. To address your comments, we conducted new experiments, performed new analyses, and have significantly revised our manuscript to lend additional support to our findings.

Note that we also made changes to some of the color plots in Figure 2 and Figure 5 to make the data more accessible for individuals with color-vision deficiencies.

Please find our point-by-point responses to the comments below:

REVIEWER COMMENTS (in italics)

Reviewer #1 (*Remarks to the Author*):

In the present study, Huerta et al. investigated the neuronal responses to immune signals generated by nodose ganglion neurons in mice. Using in vivo calcium imaging, the authors showed that nodose vagal sensory neurons display distinct responses to pro-inflammatory (TNF, IL-1 β) and anti-inflammatory (IL-10) cytokines. These responses give rise to a population coding that involves both fine-tuned and broadly-tuned neurons. Using a dextran sulfate sodium (DSS)-induced colitis model, the authors show that inflammation alters nodose ganglion neural excitability. The authors concluded that nodose ganglia neurons perform real-time cytokine encoding at relay stations upstream of the brainstem.

This paper presents a very interesting study in which the real-time responses of vagal neurons to cytokines were demonstrated. This is an important conceptual advance, as the data imply that rapid, real-time information on cytokine levels is conveyed to the brain, where counter-regulatory responses are presumably generated. Overall, the study is well-performed, timely, and well presented.

We thank you for your supportive comments on our work.

However, I am intrigued by the following. The data in Figure 1E clearly show no significant change in the amplitude of cytokine-evoked responses after the application of different concentrations of the same cytokine. However, DSS induced important changes in the excitability of nodose neurons. As DSS presumably changes cytokine levels, I expected concentration-dependent responses in nodose neurons. How did the authors interpret these results?

We appreciate the reviewer's observation and believe that it highlights an important point. We did not see evidence of different cytokine concentrations changing the neural responses, at least for the three cytokines tested in our study. We believe that this insensitivity to concentration may reflect a biological mechanism that prioritizes cytokine specificity over concentration to maintain a consistent encoding of immune signals. Therefore, if these vagal sensory neurons are activated in a threshold-like "all-or-none" manner, then further increases in cytokine concentration would not necessarily affect the

neural responses. We have now added text in the revised manuscript to discuss this point (Page 10, lines 341-343).

In the DSS-colitis model, which is a chronic-like setting of elevated inflammation, the altered excitability of these vagal sensory neurons (**R1 Figure 4A, 4B**) may reflect compensatory changes that modify baseline excitability and cytokine-specific responses through mechanisms that are independent of cytokine concentration. Similar changes in the sensitivity of colorectal afferents have been previously reported in preclinical models of colitis, providing a potential link between nociceptors and visceral hypersensitivity observed in inflammatory conditions (Feng et al., 2012 PMID: 22859364; Jain et al., 2020 PMID: 32451393; Matsumoto et al., 2012 PMID: 22330338). In the revised manuscript, we have added this information (Page 11, lines 360-366).

In addition, limited information on the transcriptomics data were provided.

We thank the reviewer for pointing this out. For the transcriptomic data, we have now performed a revised comprehensive analysis of differentially expressed genes and pathway analysis in Database for Annotation, Visualization, and Integrated Discovery (DAVID) employing an over-representation analysis (ORA) approach. Using this DAVID analysis, we now focus on changes in specific gene pathways in vagal ganglia collected from DSS-colitis mice at peak inflammation that indicate transcriptomic changes in ion channel expression, receptor sensitization, and the regulation of membrane potentials (**R1 Figure 5**). We also observed transcriptomic changes in specific genes encoding calcium channels (*Cacna1h*, *Cacna2d2*), sodium channels (*Scn10a*), and potassium channels (*Kcnk18*), which may play a role in the DSS-colitis changes in neuronal membrane potential and excitability. These additional details on the transcriptomic data, highlighting both upregulated and downregulated pathways, have now been added to the revised text (Page 7, lines 258-269) and in the Discussion section (Page 11. Lines 371-378).

Specifically, do cytokine-responsive neurons also respond to enteroendocrine hormone signals?

We thank the reviewer for this comment. Yes, cytokine-responsive neurons from the nodose ganglia likely respond to enteroendocrine hormone signals via ion channels that respond to metabolic changes occurring in the gut. Enteroendocrine cells (EECs) are known to communicate with vagal sensory neurons that innervate the gastrointestinal tract through hormones, such as cholecystokinin and peptide Y (Bai et al., 2019 PMID: 31730854; Kaelberer et al. 2018 PMID: 30237325). In the transcriptomic data, we observed an upregulation of the specific genes, including synaptotagmin-7 (*Syt7*), which has multidimensional roles in synaptic transmission, calcium-sensing, and hormone secretion. Importantly for this question, *Syt7* regulates the release of neuropeptides in response to increased intracellular calcium concentrations. We have now added text in the revised manuscript to discuss these points (Page 11, lines 371-375).

While we have not directly examined whether the same nodose ganglia neurons that respond to cytokines also respond to EEC hormones, this is a distinct possibility that we are currently investigating. Immunohistochemistry images of proximal colon tissue

from our *Vglut2-GCaMP6f* mice show that glutamatergic EECs in the myenteric plexus are located adjacent to nerve fibers that are positive for the neural marker Tuj1. An example image is shown in the figure below:

Reviewer Figure: Immunohistochemistry of the mouse proximal colon showing the close proximity of nerve fibers (Tuj1-positive) and a Vglut2-positive EEC. Work from other groups has shown that EECs release hormones to communicate with vagal sensory afferents innervating this location in the colon. Scale bar, 10 μ m.

As we did not specifically examine these EECs or their signaling in this study, we have not included the above image in the manuscript.

Are nodose neurons responsive to cytokines specifically targeting certain organs, such as the GI tract, as the DSS data suggest? How do nodose neurons sense circulating cytokines? Please comment.

We agree with the Reviewer that this is an important question about the location of cytokine sensing. To address this, we have performed additional experiments with the cytokines applied to a specific relevant end organ site, the proximal colon. In this new data (**R1 Extended Data Figure 3B**), we show that the cytokine-specificity of the neural responses is maintained when the three cytokines are individually administered within the colon. We interpret this neural activity as activating vagal afferents that innervate the proximal colon tissue. In addition, we also now provide additional immunohistochemistry of colon tissue showing that glutamatergic (Vglut2-positive) nerve fibers (Tuj1-positive), are found in the myenteric plexus, as well as the mucosal and submucosal layers of the proximal colon (**R1 Extended Data Figure 3C, 3D**).

The manuscript text has been revised to reflect this new data (Page 4, lines 157-161). Additional text discussing potential mechanisms for cytokine sensing have also been added to the Discussion (Page 10, lines 352-354).

Reviewer #2 (Remarks to the Author):

The authors demonstrated a real-time neural response to pro- and anti-inflammatory cytokines within the nodose ganglia and their selectivity of responses to these cytokines. The authors also showed that neurons in nodose ganglia are activated but their signaling capacity to specific cytokine was decreased under DSS-induced colitis. This study is very important in that vagal sensory neurons could encode cytokines as neural signals under disease. However, there still remains unclear how encoding cytokine-dependent signaling in VG neurons could impact systemic diseases in VG-dominated organs. Here are some comments from the reviewer.

We thank the reviewer for the positive comments on our work.

#1. The authors have shown that vagal nerves respond to cytokine stimulation in terms of DFF amplitude and the pattern of DFF differed depending on the types of cytokines. However, it is vague what functional outcomes could be brought about after cytokine stimulation onto the vagal nerves. Without this functional assay, it is unclear how responses to cytokines in vagal nerve could impact on DSS-induced colitis. One suggestion could be evaluating the production of neuropeptides after cytokine exposure onto the vagal nerve and also testing if there is any cytokine-dependent differences.

We thank the reviewer for these suggestions and agree that a functional outcome would be an important aspect. While did not directly test whether cytokine exposure to the vagus nerve changed any DSS-colitis outcomes, we would like to point out that the nodose ganglia functional activity (baseline DFF) remained decreased at Day 14 despite a resolution of most DSS-colitis symptoms (**R1 Figure 4E**). This decreased DFF correlated with a decreased colon length at that timepoint (**R1 Figure 4F**), suggesting that vagal sensory neuron activity at baseline may be a more reliable indicator of the internal organ state, such as colon tissue status, than observation-based clinical or behavioral scoring.

Regarding neuropeptides, in the revised manuscript text and **R1 Figure 5**, we highlight the transcriptomic findings that *Ramp1* and *Ramp2* are decreased in RNA seq of DSS-colitis ganglia, potentially indicating changes in signaling for the neuropeptide calcitonin gene-related peptide (CGRP). Neuropeptides, such as CGRP and neuropeptide Y, are known to be released by sensory neurons to interact with RAMP1 on intestinal EECs to maintain the gut barrier and protect against inflammation (Chu et al., 2020 PMID: 32187517). Prior work has shown that neuropeptides, including substance P and CGRP, are released by sensory neurons following activation by cytokines (Chu et al., 2020 PMID: 32187517; Crosson et al., 2021 PMID: 33453289; Miller et al., 2009 PMID: 19655114). The manuscript text has been revised to discuss this important point about the role of neuropeptides (Page 10, lines 352-354 and Page 11, lines 366-370).

#2. The author found more active nodose ganglia neurons under DSS-condition at day7 (Figure 4B). In the meantime, they also found that these nodose ganglia neurons are less sensitive to cytokine stimulation. The authors explain this discrepancy could reflect the desensitization of nodose ganglia neurons. However, a body of studies have proven protective roles of VG in reducing systemic inflammation including DSS-induced colitis. If VG becomes desensitized during DSS colitis and less responsive to cytokines, how could VG modulate disease through vagus nerves stimulation.

The reviewer is correct that vagus nerve stimulation (VNS) has been shown to be beneficial in both preclinical and clinical settings for reducing systemic inflammation in inflammatory bowel diseases. VNS to reduce systemic inflammation occurs through the motor efferent arm of the inflammatory reflex, sometimes termed the cholinergic anti-inflammatory pathway. Work from our lab and others has shown that activation of the cholinergic motor efferent vagal fibers, or their origin in the brainstem dorsal motor nucleus of the vagus, reduces levels of circulating cytokines and inflammation (Chavan

et al., 2017 PMID ; Kressel et al., 2020 PMID ; Tracey, 2022 PMID:). Recent work has shown that VNS may also trigger gut- and microbiome-specific changes that are independent of reductions in systemic inflammation (Chang et al., 2024 PMID: 39121857).

However, in this study, we have focused on the sensory afferent pathway and have not examined the effect of VNS specifically. In the setting of inflammation, vagal tone is decreased, as shown by the loss of high-frequency heart rate variability (parasympathetic) and increase in low-frequency variability (attributed to an excess sympathetic stimulation). For example, the autonomic dysfunction in rheumatoid arthritis patients is characterized by an increased overall sympathetic tone and decreased activity of the vagus nerve, as measured by vagal tone (Koopman et al., 2011 PMID: 21607292). This explains why endogenous vagal activity in the setting of inflammation provides much less anti-inflammatory activity than direct electrical stimulation (vagusotomy shows that there is some anti-inflammatory activity, but it is typically less than direct electrical stimulation). This inflammation-associated reduction in vagal activity may explain, at least in part, the reduced baseline amplitudes we observed in the DSS-colitis condition (**R1 Figure 4C**). Additionally, the increased number of active neurons is consistent with the existence of an inflammatory milieu in which increased concentrations of cytokines could recruit more active neurons (**R1 Figure 4A, 4B**). Neuronal desensitization via receptor or post-receptor modulation is typically a feature of sustained, strong molecular stimulation. Whether this phenomenon explains our experimental observations will require additional experimental work.

The manuscript text has been revised to include these points, along with supporting references (Page 11, lines 360-366).

#3. The authors showed neural responses to IL-1 β and TNF α in Figure.2 but could they also show any data on the stimulation with IL-10 and IL-10R expression in VG? The reviewer is interested in IL-10 data because the authors have shown that VG neurons are more sensitive to IL-10 compared to other cytokines according to DFF amplitude (Figure.1 D,E). Do IL-10 responsive (or IL-10R expression) subset of nodose neurons possess more distinctive features?

We did not perform the cytokine combination experiments with IL-10 application, but the reviewer brings up an important point. While the IL-10 elicited neural responses are higher in DFF amplitude, compared to TNF and IL-1, we do not know what accounts for this and whether this is associated with varied IL-10R expression on individual nodose ganglia neurons or due to differences in transduction of these anti-inflammatory signals. This is the subject of additional ongoing experiments.

We did show that there is expression of IL10R on the vagal ganglia and on the vagus nerve itself (**R1 Extended Data Figure 4**). Being a canonical anti-inflammatory cytokine, it is possible that these VG neurons encode and transduce IL-10 signals in a different manner than those for proinflammatory signals (TNF and IL-1). We will explore this possibility in future studies.

#4. Colon is innervated from DRGs as well as VG, and DRGs could also encode cytokine signals. Then, to what extent vagal cytokine sensing is involved in this study?

Are enteric DRG neurons could also show similar response to IL-1 β , TNF α and IL-10?

Yes, the colon is innervated by both vagal afferents and DRGs. It is entirely possible, and likely, that DRGs also encode similar cytokine signals that modulate both physiological and neuro-immune responses. While we did not examine DRGs or enteric sensory neurons in this study, work by other lab groups has shown neuronal sensitization from inflammatory cytokines and direct activation by certain cytokines on DRGs (Oetjen et al., 2018; PMID: 28890086; Yang et al., 2023 PMID: 36947942). We have added text to highlight the Reviewer's point about a potential role for DRGs in this neuro-immune signaling (Page 11, lines 388-390).

To determine whether the cytokine-elicited nodose ganglia responses we examined can also be triggered by stimuli within the colon, we have performed additional calcium imaging experiments where we administered the three cytokines in the proximal colon (**R1 Extended Data Figure 3B-D**). The mouse proximal colon is innervated by Phox2b-positive glutamatergic vagal sensory neurons, while DRGs express *Prdm12* and are known to mostly innervate the mid and distal colon (Feng et al., 2012 PMID: ; Guo et al., 2021 PMID: 33533318). Given this innervation pattern, we have now added additional experimental data to show that the nodose ganglia neurons that we are studying also encode cytokine-specific signals from this innervated end organ.

Reviewer #3 (Remarks to the Author):

Huerta used in vivo imaging of nodose ganglia in Vglut2-GCaMP6f mice to investigate the effects of pro-inflammatory cytokines (TNF, IL-1 β) and the anti-inflammatory cytokine IL-10 on neuronal responses in the nodose ganglia. Their findings suggest that some nodose ganglia neurons exhibit cytokine-selective responses, as indicated by increased GCaMP6f signals. Furthermore, the authors observed cytokine-specific patterns of activity, though the biological significance of these diverse responses remains unclear.

The study also assessed neuronal activity in the nodose ganglia using a dextran sulfate sodium (DSS)-induced colitis model. DSS-induced inflammation increased the number of active nodose ganglia neurons at baseline but reduced their signaling capacity and responsiveness to specific cytokines, indicating altered neural excitability.

Transcriptomic analysis of vagal ganglia from DSS-treated mice revealed a down-regulation of cytokine signaling pathways, alongside an up-regulation of neuronal activity pathways. However, the mechanistic connections between these transcriptomic changes and functional outcomes were not investigate, limiting the scope of this study to an observational study, due to limited mechanistic insight, leaving the broader significance of the findings unresolved.

We thank the Reviewer for these important comments, which we have addressed in this revision with additional data and analyses to link the transcriptomic and functional results, with a focus on potential mechanistic insights. Additional specific responses to the Reviewer's comments follow below.

A primary concern for this reviewer is the reliability of the detected signals. Careful

examination of the supplemental movie and the representative image in Fig. 1 suggests a low detection threshold. On page 4, the authors state that cytokine concentration influenced response profiles, yet they also imply that the cytokine-specific responses were insensitive to concentration. This inconsistency raises concerns. Based on Supplemental Movie 1, the low detection threshold—likely due to a low signal-to-noise ratio—may have compromised the accuracy of the results and, consequently, the interpretation of the data.

Thank you for your careful examination of the data. The image in the original Figure 1B was of the raw fluorescence signal in the field-of-view of the Miniscope, it was not a DFF image. In the revised manuscript and **R1 Figure 1B**, we have replaced the raw image with an improved one and added a representative DFF image next to it for clarification. We have also added additional details clarifying this in the revised Figure Legend. We apologize for the initial confusion.

We do not see where we stated that cytokine concentration influenced response profiles in the original manuscript. If that was implied, then it was a mistake, and we hope that additional text clarifications (Page 10, lines 341-343) and our revised dose-response data (**R1 Extended Data Figure 3A**) now make it clear that we did not observe concentration-dependent changes in this study.

Additional comments: In Fig 1D the signal amplitude for IL1B, TNF and IL10 on average are around 25, 50 and 75, but for the dose response experiment in 1E the average signal amplitude is between 50 to 60 – what dose they used for D? why the differences are gone? Are the Ns relating to the activated number of cells or combined # of slice plus activate cell number

We thank the Reviewer for pointing this out. The concentrations used for experiments in the original Figure 1D were IL-1 β , 200 ng/mL; TNF, 50 ng/mL, and IL-10, 50 ng/mL, which are the middle doses shown on the dose-response data (now **R1 Extended Data Figure 3A**). This information has now been added to the Figure Legends (Page 13, lines 434-435).

The sample sizes and data points listed in the original **Figure 1E** were of individual neuronal responses. In the revised manuscript, these dose-response results have been moved to **R1 Extended Data Figure 3A**, and we have included the responses as mean values per mouse to match the data in the other plots. The details on the number of responses, the number of mice, and the concentrations used have all been added to the **R1 Extended Data Figure 3** and accompanying Figure Legend (Page 15, lines 517-519). Thank you for noticing this, as we agree that these details are important to specify.

Please include sample images for each condition- it is very important to see the images before they are processed for analysis.

Thank you for your comments on this. We have included examples of the raw data for each of the three cytokines tested below, before they are processed for analysis. These representative examples show that we have a signal-to-noise ratio and a detection threshold that allows for reliable discrimination of responses over the baseline activity.

Reviewer Figure. Raw data images from the Miniscope are shown on the top row, with example DFF frames shown below of a cytokine-responsive neuron for each cytokine tested. Note that a single frame is shown in the DFF images to highlight the detection of a single responsive ROI.

What is the mechanistic link between Data shown in Fig 5 (Transcriptomic analysis of DSS-colitis vagal ganglia), and data shown in Figure 1-4?

Thank you for this comment. In the revised manuscript, we have now performed a revised comprehensive analysis of the transcriptomic data, including differentially expressed genes (DEGs) and pathway analysis in Database for Annotation, Visualization, and Integrated Discovery (DAVID) employing an over-representation analysis (ORA) approach. In the original manuscript, we used Gene Set Enrichment Analysis (GSEA) to evaluate pathway enrichment across the entire ranked gene list without applying significance thresholds. While GSEA did identify biologically meaningful pathways, the false discovery rates (FDRs) may have been too high due to the inclusion of non-significant genes, reducing confidence in the results. Now in the revised manuscript, following consultation with our geneticist colleagues, we use a DAVID-based ORA approach that focuses exclusively on genes meeting specific statistical thresholds. By narrowing the analysis to this subset of genes, ORA reduces the impact of variability in less significant genes, enabling the identification of pathways with lower FDRs and more robust statistical support. The manuscript text has been revised to reflect these new details and references added for the DAVID approach (Page 28, lines 855-870).

In the revised manuscript, we have added further explanation of the potential mechanistic links between the transcriptomic data and functional data. Using this DAVID analysis, we observed changes in specific gene pathways related to ion channel expression, receptor sensitization, and regulation of membrane potentials (**R1 Figure 5**). We also observed transcriptomic changes in specific genes encoding calcium channels (Cacna1h, Cacna2d2), sodium channels (Scn10a), and potassium channels (Kcnk18), which may play a role in the DSS-colitis changes in neuronal membrane potential and excitability.

These additional details on the transcriptomic data, highlighting both upregulated and downregulated pathways that may contribute to the functional changes we observe in DSS-induced colitis, have now been added to the revised Results text (Page 7, lines 258-269) and Discussion (Page 11, lines 366-378).

REVIEWERS' COMMENTS

Reviewer #1 (Remarks to the Author):

This is an improved version of an elegant manuscript reporting on neuronal sensing mechanisms of immune signals. The authors have in my view addressed the concerns to satisfaction, so that the present version constitutes an important and timely contribution to the field.

We thank you for the supportive comments and input.

Reviewer #2 (Remarks to the Author):

The authors have sufficiently addressed all concerns.

We thank you for your input.

Reviewer #3 (Remarks to the Author):

The signal-to-noise ratio of the best representative raw data for each of the three cytokines tested further supports a near detection limit for the Miniscope images before and after IL1B, TNF and IL10. Aside from this concern, the authors have adequately revised the manuscript and addressed this reviewer's comments. I enjoyed reading this paper.

We appreciate your close examination of the data and input on the work. Thank you for your positive feedback.